# Promoting the Sustainable Recovery of Hospitality in the Post-Pandemic Era: A Comparative Study to Optimize the Servicescapes

**DOI:** 10.3390/ijerph20021100

**Published:** 2023-01-08

**Authors:** Maria M. Serrano-Baena, Rafael E. Hidalgo Fernández, Carlos Ruiz-Díaz, Paula Triviño-Tarradas

**Affiliations:** 1Department of Graphic Engineering and Geomatics, Campus de Rabanales, University of Cordoba, 14071 Cordoba, Spain; 2Department of Architectural Constructions I, University of Seville, 2, De la Reina Mercedes Ave, 41012 Seville, Spain

**Keywords:** hotel, servicescape, post-pandemic, resilience, sustainability, recovery

## Abstract

As COVID-19 spread throughout the world, the hospitality and tourism sectors were hard hit as no other industry. For this reason, the UNWTO developed the One Planet Vision as a response to a sustainable recovery of the tourism sector. At present, when people are starting to travel and stay at hotels again, it is important to analyze what their expectations are of hotels to move forward in the post-pandemic era. For instance, empirical research has been developed to examine people’s sentiments toward servicescapes, and a comparative study is presented between 2020 and 2022. Findings contribute to the research by identifying new servicescape attributes during a health crisis. These also lead to practical implications by proposing a scale to evaluate customers’ perceptions and to increase their wellbeing and resilience. The current research is one of the first studies to collaborate with the One Planet Vision by empirically proposing improvements in the servicescapes of hotels for a responsible recovery.

## 1. Introduction

Since the beginning of the pandemic, the tourism and hospitality sectors have been dramatically disrupted and forced to make adaptations. With the introduction of social distancing and traveling restrictions as the most common responses to contain the virus, the tourism and hospitality industries were, as no other, hard hit. Due to its negative effects, researchers have studied its implications throughout the different stages [1,2]. Nowadays, most of the relevant studies are focused on its socio-economic impacts but there is limited work reviewing the resilience aspect [3,4]. Moreover, a small number of researchers have examined the perceptions and emotions of individuals, both customers [5] and employees [6,7], during the pandemic to gain insights into people’s behavioral responses. It has been concluded that understanding people’s emotions is crucial to successfully operating in the post-pandemic era and building their resilience. However, the existing literature about the social dimension of sustainability which includes wellbeing, working conditions, and human rights, among others, is limited [8].

In a sustainable context, some existing studies have investigated the sustainability in the hospitality sector [9,10]; however, the current health crisis might entail new ways of approaching the 2050 neutrality proposed by the European Union and other relevant future sustainable plans. With the aim of successfully achieving these, some further and deeper research should be conducted about how the sentiments of people have progressed from the beginning of the pandemic to the current post-pandemic period in a hotel context; additionally, it should also be studied how the different servicescapes could be sustainably improved to recover from the adverse effects of an airborne health crisis, such as COVID-19. To address this gap and sense of urgency, this paper studies the negative impact the pandemic has had on the hospitality industry by analyzing how the sentiments of people have changed toward hotel servicescapes and what they expect to find in them from now on. The following questions have guided the research:-RQ1. Which are the servicescape attributes and dimensions to consider during an airborne health crisis, such as COVID-19?-RQ2. How have the sentiments of people toward the servicescapes of hotels changed from 2020 to the post-pandemic era?-RQ3. What are the main attributes that must be taken into account for a sustainable recovery of hotels in the post-pandemic era?

Empirical research has been developed in this paper to collaborate with the One Planet Vision proposed by the UNWTO as a response to a sustainable recovery in the tourism and hospitality sectors [11]. A comparative study between people’s opinions about servicescapes in November 2020 and September 2022 has been developed in the tourist region of Andalusia, Spain. The novelty of this paper lies in the recent status of the post-pandemic stage and the lack of empirical research proposing improvements in hotels for a responsible recovery toward 2050 neutrality.

## 2. Literature Review

### 2.1. Implications of COVID-19 in the Sustainable Development of Hospitality

The COVID-19 pandemic has caused major negative consequences on a global scale. The UNWTO Secretary-General Zurab Pololikashvili has expressed that “COVID-19 has impacted travel and tourism like no other event before in history. Governments have put public health first and introduced full or partial restrictions on travel. With tourism suspended, the benefits the sector brings are under threat, millions of jobs could be lost, and progress made in the fields of equality and sustainable economic growth could be rolled back” [12]. Several studies have provided a review of the health crisis and its effects on the tourism sector. The factors that affect the recovery of tourist destinations were analyzed and it was concluded that the effects will differ in space and time [13]. Some places might continue their businesses as usual ignoring the possibility of a new era of green growth, but others may reconsider the reorientation of their businesses following a sustainable approach [14]. As it has been found, the pandemic could motivate stakeholders to think more sustainably, by fighting for an advantage in the post-pandemic stage [15].

The UNWTO has expressed the importance of tourism in the Sustainable Development Goals of the 2030 Agenda. The Agenda contemplates tourism specifically in Goals 8, 12 and 14. It expresses the need to implement new policies that promote sustainable tourism, adopt more responsible production and consumption strategies, preserve ecosystems, and promote a blue economy [16]. It has also highlighted the need to reduce the current emissions of tourism by 50% before 2030 in order to achieve the goal of CO_2_ neutrality by 2050. It is in this context, in May 2020, when the UNWTO proposed the One Planet Vision. This plan was created as a recovery response to help the tourism and hospitality sectors to emerge stronger, rebuild sustainably, and improve their resilience after the pandemic. It establishes six axes of action: climate action, the circular economy, the conservation of biological diversity, social inclusion, public health, and governance and finance. In addition, the Glasgow Declaration on Climate Action in Tourism arises from the need to accelerate climate action in tourism and ensure strategies that help achieve the 2050 neutrality goals established by the European Green Deal [17]. Considering that hotels are included in the top five list of energy consumers in the tertiary sector [18], this makes them an exceptional goal to be ecologically improved. Therefore, in order to achieve the goal of CO_2_ neutrality, it is necessary to involve hotels in the process and include updated measures for their recovery [9]. Regardless of the newness of this topic, some researchers are now evaluating the effects of the pandemic on sustainable development [1,2,19,20,21]. For instance, Elkhwesky et al. (2022) developed a comprehensive literature review of sustainable practices in hospitality and demonstrated important progress from 2020 to 2021. Renzi et al. (2022) evaluated consumers’ awareness of the achievement of Agenda 2030 in terms of sustainable behavior after COVID-19. Moreover, Gössling and Schweiggart (2022) reviewed the existing literature published about tourism and COVID-19; they suggested the importance of studying the pandemic to gain some insights into the management of climate change. Waste management in the hospitality sector has also been evaluated in the post-pandemic era. Filimonau (2021) proposed potential strategies to address food and plastic waste; furthermore, the author highlights the need of investing in green strategies and innovation for a sustainable recovery. The current period, when people are starting to feel more comfortable traveling, is an opportunity to strengthen the hospitality and tourism sectors by sustainably improving their buildings. 

### 2.2. Hospitality Servicescapes Effects

Widespread research has been published about the servicescape effects on hotel customers’ and employees’ conduct [22,23,24]. Bitner (1992) proposed the concept of “servicescape”, one of the most widely cited conceptual frameworks, by exploring the impact that physical surroundings have on customers’ and employees’ behaviors. The servicescapes are constituted by a mix of environmental features that influence internal responses and behaviors. These include all the objective physical factors that the firm can control in order to enhance or constrain the customer and employee actions. These factors comprehend an extensive list of elements, such as layout, materials, lighting, temperature, signage, and color. Bitner identified three primary dimensions of the servicescape: ‘Ambient conditions’, ‘Spatial layout and Functionality’, and ‘Signs, Symbols and Artefacts’ that influence people’s perceptions. The second and third were commonly referred to as the ‘Built Environment’. Later on, Wakefield and Blodgett (1994) studied the importance of servicescapes in leisure and focalized on the ‘Built Environment’. Their findings suggest that the servicescape has a powerful effect on the time that customers wish to stay in the leisure service and is a determinant for customers’ intentions.

In a more recent context, a comprehensive literature review has revealed a number of studies on the dimensions of the servicescapes in the hospitality industry [25,26,27,28,29,30,31]. Spielmann et al. (2012) proposed a scale incorporating physical complexity and social interaction. The scale indicated how certain perceptions influenced consumers’ behaviors. Lockwood and Pyun (2018) reviewed the attributes and dimensions of the hotel servicescape and revealed five main factors with high reliability, these are ‘Aesthetic Quality’, ‘Functionality’, ‘Atmosphere’, ‘Spaciousness’, and ‘Physiological Conditions’. In 2019, they also tested this scale and proved it to be valid and reliable. Their study showed that four of the five dimensions contribute to the feelings of pleasure and arousal; the effect of ‘Functionality’ was not confirmed, and this might be due to the characteristics of the hotels selected for the survey. Contrarily, Wakefield and Blodgett (1994) proved that ‘Functionality’ plays a very important role in the leisure service, so its effects have been proved to be also a determinant. Ozkul et al. (2019) explained the importance of lighting and color in the service atmosphere in tourism and hospitality for the customer’s perception and satisfaction. The methodology proposed some design recommendations for color and lighting for different services. Trinch (2021) evaluated the existing literature review about the impact of COVID-19 on the hotel service industry. As a result, it was concluded that addressing the weaknesses and transforming these into opportunities will solidify the resilience of the hospitality sector. Moreover, Willems et al. (2021) demonstrated how the COVID-19 crisis could prompt the digital and technological boost of services and retailing. Finally, Lugosi et al. (2022) examined the effects of several servicescape dimensions on visitors’ satisfaction in a cancer treatment context. The study considered design, spatial layout, functionality, ambient conditions, and physical surroundings, among others. Findings showed that ambient conditions had the greatest impact on satisfaction. It was also found that the right management of these dimensions might help to compensate for some deficiencies in other areas of the servicescape, especially during the the COVID-19 pandemic.

## 3. Materials and Methods

In order to carry out this empirical research, the methodology was developed into three phases: 

(I) The generation of a proposed servicescape scale in a health crisis environment. This or similar events have not been studied before from a servicescape perspective and the previous scales did not reflect any attributes related to an airborne health crisis, such as COVID-19. Hence, due to the novelty of this topic, the authors created the proposed scale shown in Table 1 based on the existing literature related to servicescapes and the recent research about the pandemic.

(II) An online administered questionnaire was designed from the proposed servicescape scale. Participants were asked to answer the questionnaire from a pandemic perspective and consider the context they were in at the time of responding. The reliability of the questionnaire and the margin of error were studied to validate the survey statistically. In this case, convenience sampling was considered to be the best technique for the survey. This type of non-probability sampling involves the sample being drawn from a part of the population that is close to hand. 

(III) Subsequently, a quantitative method was used and information was collected from the questionnaire. The survey was conducted at two different stages: the first one during the high peak of the disease in November 2020 and the second one in September 2022, once the restrictions were eased. An analysis and comparison of results were performed in order to provide insights about how people’s sentiments have changed over the time of this pandemic and what is expected from now on in a hotel.

### 3.1. Proposed Servicescape Scale for an airborne Health Crisis

Table 1 shows the authors’ proposed scale. This scale is the result of the most relevant dimensions exposed by previous authors about servicescapes [22,28,32,33], together with the latest research on airborne transmission diseases, the current COVID-19 pandemic, and its implications in the servicescape. For this particular study and future research involving the servicescape during and following the pandemic, the authors have created an upgraded scale with further attributes to be considered and have added a proposed sixth dimension under the category of ‘Services’, including the main services affected in the hotel during the current pandemic that are crucial for this context [34,35,36,37,38,39,40,41,42,43,44,45,46,47,48,49,50,51,52,53,54,55,56,57,58]. Hence, the first column of Table 1 shows the proposed 6 main dimensions of the proposed servicescape scale. The second column shows the proposed attributes related to each dimension; the attributes shown with a * have been proposed by the authors based on the existing research in a pandemic context, and the rest of the attributes have been selected from previous authors. The third column corresponds to the servicescape references that support the proposed attributes and the fourth column corresponds to the COVID-19 references. All of the attributes are supported by existing research for a pandemic context, as shown in the fourth column. Due to the lack of previous research that validates the relationship between the disease and some of the existing attributes that other authors proposed in the past, they have been omitted from the proposed scale. These are ‘The height of tables and chairs’, ‘The design of the hotel’s exterior’, ‘The fabrics used’, and ‘Texture/pattern’.

### 3.2. Questionnaire Design and Reliability of the Survey

The questionnaire is based on a five-point Likert-type scale to express importance (1. unimportant to 5. very important), quality (1. very poor to 5. excellent), agreement (1. strongly disagree to 5. strongly agree), or satisfaction (1. very dissatisfied to 5. very satisfied). A total amount of 31 questions or statements grouped into 6 sections were provided, each of them related to one or more attributes proposed by the authors in Table 1. An online approach was determined to be the best procedure, distributed via Google Forms; the first approach in November 2020 and the second approach in September 2022. The first group was formed with a total of 223, including 222 valid responses. The second group was formed with a total of 249, including 241 valid responses, comprising all participants aged between 18 and 90 years old of all genders based in Andalusia. According to the Statistic National Institute of Spain, the population comprised between 18 and 90 years old in Andalusia in 2020 was 5,916,787 people [59]. The margin of error (MOE) was calculated following the following formula: MOE = z ∗ √p ∗ (1 − p)/√(N − 1) ∗ n/(N − n)(1)
where z = 1.96 for a confidence level (α) of 95%, p = proportion (expressed as a decimal), N = population size, and n = sample size.
MOE _Andalusia 2020_ = 0.98/14.9 ∗ 100 = 6.57%MOE _Andalusia 2022_ = 0.98/15.5 ∗ 100 = 6.31%

Hence, with a 95% of confidence level, the MOE for Andalusia in the 2020 survey is ±6.57% and ±6.31% in the 2022 survey. In order to check the reliability of the questionnaire, Cronbach’s alpha coefficient was calculated to measure the internal consistency of the 31 questions. In this case, the internal consistency was acceptable, with Cronbach’s alpha being α = 0.70 in both cases. The significance level for this research is 0.05. 

## 4. Results and Data Analysis

The raw data from the survey were extracted in the shape of an Excel file containing all the questions and the Likert scores from each individual. In order to statistically study the significance of each dimension, a t-test has been conducted to analyze the quantitative data extracted [60]. A *p*-value ≤ 0.05 proves that the null hypothesis is rejected, showing a difference in the sentiments of people from 2020 to 2022. Contrarily, a *p*-value > 0.05 proves that there is no significant effect and the null hypothesis must be retained in those cases, showing that people’s sentiments have not significantly changed. The analysis of the survey is divided into two steps with the purpose of fulfilling a complete data analysis. The first step consists of statistically interpreting the interval data; this includes the mode, mean, standard deviation, *p*-value, and t-value for each dimension and period of time. The second step is to analyze the data by interpreting the charts extracted from each category of the survey. The descriptive statistics for each of the dimensions and attributes are presented below. As a means to obtain a full understanding of respondents’ behaviors and thoughts, they were asked a series of questions regarding their perception of the servicescape dimensions of the hotel during COVID-19 and at the post-pandemic stage, and their responses are shown below.

### 4.1. Analysis of the Aesthetic Quality Dimension

An analysis of people’s responses toward the aesthetic quality dimension of the hotel, see Table 2 and Figure 1, show that the pictures and photos on display do not affect their wellbeing or make a positive or negative impact on their resilience. This finding is supported by the mode and mean from Q1, which highlight the neutrality of this attribute. The key finding of Q2 suggests that the use of flowers and plants in the hotel has a positive impact on people. This has not changed throughout the health crisis and it seems to be as important in 2022 as it was in 2020. However, the addition of plants within the room does not seem to affect people’s perception of the quality. This can be extracted from the mode of Q3, where the majority of respondents found the room acceptable without any plants. These results have not changed in the post-pandemic stage, so it can be assumed that people feel the same way about this attribute independently of the stage of the pandemic they were in. On the other hand, the style/design of the ornaments, furniture, and flooring positively affects their resilience toward COVID-19. People gravitated toward a more minimalistic style and a clean-lined design seemed to improve the quality of the space from people’s perspective, making them feel more comfortable in it. This can be extracted from the mode and mean of Q4 and Q5 which have remained constant. 

Figure 2 shows the percentage of the responses for each of the five-point Likert scale scores of the aesthetic quality dimension. Results of Q1 show that about 40% of the respondents claimed that the use of artwork was unimportant or slightly important for their comfort in the hotel during both stages. In this case, respondents did not have a resounding answer. Similarly, in Q2 for approximately 60% of them, it was important and very important to find plants in the hotel, and 30% expressed that it was moderately important in both stages of the pandemic. Indoor plants in the room did not affect their perception of the quality for 55% of people, as shown in Q3, where the room was found good or excellent without them. In regards to Q4 and Q5, answers are very similar but the responses changed slightly from 2020 to 2022. In the first stage, more than 85% of the respondents expressed that they felt very comfortable with a minimalistic decoration style and find a room with clean lines of furniture of excellent or good quality. Responses from Q4 have lessened by 5% in 2022, while responses from Q5 show an increase of 3% from the first stage. From this section of the questionnaire, it can be extracted that people’s sentiments toward the aesthetic quality of the hotel have not dramatically changed in the past two years and their responses have remained constant.

### 4.2. Analysis of the Functionality Dimension

Some interesting findings can be extracted from Table 3 and Figure 3. Q6 shows a unanimous response regarding the implementation of physical distancing. This can be extracted from the standard deviation below in both stages of the survey. Similarly, Q8 also shows a unanimous response although, in this case, the mode has been reduced from 5 to 4 in the second stage. On the other hand, Q7 and Q9 present a varied range of responses, especially in the first stage. Q7 shows that people have slightly changed their minds and, in 2022, they only disagree with the asked statement. It can be interpreted that people did not strongly agree with distancing furniture; this is probably reinforced due to the post-pandemic stage and resistance against the virus developed within the last two years. Q9 remains similar and people still disagree with the use of banners affecting their comfort. In short, the mode from Q6 and Q8 shows the high importance and support that people give to the use of masks and physical distancing with resounding responses, as shown in the SD. Conversely, Q7 and Q9 are more dispersed. Their SD values show us the spread of these responses.

Results of Q6 in Figure 4 shows that in 2020, 97% of the respondents found it important and very important the implementation of physical distancing in the hotel. This percentage has been slightly reduced to 87% in the 2022 respondents. On the other hand, the results of Q7 were quite diverse for both stages of the survey. In 2020, 50% of the participants did not agree with the stated action making them uncomfortable, while the other 50% of people thought the opposite. In 2022, the percentage of people that did not agree with the statement increased to 54%. From the first stage to the second stage, it can be noticed that people’s sentiments tend to concentrate toward more neutral responses while avoiding extreme answers. The categorical answers of Q8 in 2020 reflect that only 2% of the respondents disagreed with the use of masks in the hotel. Additionally, 77% of them strongly agreed with their use. However, these responses have been reduced to only 34% in 2022. Lastly, for Q9, 35% of the people could confirm that the use of floor stickers, signs, and banners during COVID-19 might affect their comfort while in 2020, 53% established that these measures did not affect it. Results from 2022 are similar and 28% of respondents confirmed that these measures might affect their comfort. 

### 4.3. Analysis of the Atmosphere Dimension

The results found in the atmosphere dimension are consistent throughout both stages; see Table 4 and Figure 5. From Q10 and Q11, the mode reflects the importance to perceive a quiet and relaxing background. While in 2020 this attribute was very important in 2022, the majority of people find it only slightly important. In addition, chill-out music and a quiet environment could also help with their wellbeing, as can be seen in the mean in Q11 of both stages. The use of warm lighting seems to make people feel calmer. The mean and mode for Q12 show that people agreed with the use of warm artificial lighting, and this aspect remained almost the same in both stages. Similarly, with respect to the amount of natural light mentioned in Q14 and Q15, a vast majority of the respondents felt the amount of natural light very important, in this case in the lobby and restaurant both in 2020 and 2022. The SD being ≤1 reflects the undispersed responses. As for the Q13, people were also very strong in their responses and most of them found the attribute of Cleanliness; very important. This is clearly shown in the mode and the mean in both stages.

As shown in Figure 6, Q10 reveals that for 76% of the respondents, it was important or very important to have a quiet environment in 2020, and for 74% of them in 2022; so, responses remained very similar in the two years studied. Q11 states that chill-out music would make them feel more comfortable, and 65% of the participants agreed with this statement in the first stage and 73% in 2022. In both stages for Q12, about 88% of the respondents expressed that the use of warm light in their guestrooms makes them feel calmer. The results for Q13 provide clear evidence that the respondents found it important, 25%, and very important, 67%, to perceive a clean scent within the hotel during the pandemic. Only 1% did not find this important and 6% moderately important. Very similar responses have been reported in the post-pandemic phase. Q14 and Q15 asked about the importance of natural light in the lobby and restaurant, respectively. Both questions obtained similar results in 2020 and 2022; more than 85% of the people found the use of natural light important and very important. Surprisingly, these last attributes have not changed in two years, so it can be assumed that natural light is a very important aspect of the servicescape that have not been altered since COVID-19.

### 4.4. Analysis of the Spaciousness Dimension

Based on the results of the questionnaire, see Table 5 and Figure 7, the feeling of spaciousness, free space, and outdoor/indoor areas have been clearly highlighted as positive attributes for people’s resilience during and following the pandemic. Q16 and Q17 are congruent with each other and reveal that people prefer a big guestroom over a big bathroom for both stages of the survey; this can be noticed by comparing the mode of Q16 to the mode of Q17. Moreover, Q19 and Q20 responses reflect the necessity of outdoor areas and an indoor patio to improve people’s wellbeing. Again, responses regarding these attributes have remained constant. Specifically, the use of outdoor areas is very important for a vast majority of the respondent; the mode of Q19 supports this finding. On the other hand, Q18 reflects that a double-height space in the lobby does not seem to be relevant for a considerable amount of people.

Figure 8 shows the percentage of the responses for each of the five-point Likert scale scores of the spaciousness dimension. The items in Q16 and Q17 examine the designated space for the bathroom and guestroom. In Q16, very similar results were extracted and 69% of people preferred a big guestroom and a small bathroom, while 22% in 2020 and 18% in 2022 were neutral about this statement. Conversely, results from Q17 revealed that only 15% of people agreed about preferring a big bathroom and a small guestroom in 2020, and 19% of them in 2022. Results of Q18 are almost identical in both stages and showed that for approximately 18% of people, it was important or very important to have a double-height space in the lobby, and 53% revealed that this was not important for them. In 2020, 87% of the respondents of Q19 expressed a great importance to find an outdoor common area in the hotel. This number was slightly reduced in 2022, with 85% of people expressing a great importance. Similarly, Q20 reveals that 76% of the participants from the first stage agreed with the statement that ‘an indoor patio would make you feel more relax and less overwhelmed’, and 81% agreed to this in 2022. Only less than 8% of the respondents disagreed with it in both stages of the survey. 

### 4.5. Analysis of the Physiological Conditions Dimension

The findings in this section are consistent with the results in the previous ones; see Table 6 and Figure 9. All the questions regarding the physiological conditions seem to show a positive general response. In this case, the use of regular hygiene standards and sanitation facilities is supported by people, as can be seen in the mode and mean in all questions. Participants are willing to cooperate with these measures and consider them important to reduce the spread of the virus. While in Q7 and Q9, some dispersed results were found and people felt hesitant when the measures affected their comfort, in this case, regarding the hygiene standards, all the responses are almost unanimous, as can be extracted from the SD which is ≤1 in all questions. Moreover, another important factor can be extracted from Q24 with respect to the air quality. Although all responses seem to be positive, there has been a slight drop in regard to Q21, Q22, and Q25 in 2022. Responses to the three questions showed a mode of 5 in 2020 and this has been reduced to 4 in 2022. For instance, people seem to be less satisfied with constant reminders about the virus, such as a signage of hygiene standards.

Figure 10 proves the consistency of this section. The results seem to be very positive for all five questions in both stages but all of them show a slight decrease in 2022. For instance, Q21 reveals that 96% of the respondents in 2020 were satisfied with hotels’ need of providing regular hygiene reminders. This amount, although still high, has been reduced to 90% in 2022. Q22 shows that for 91% of the people from the first stage of the survey, it was important or very important to stay in a hotel with enough amount of sanitation facilities; while in the second stage, it was important for 88%. Q23 asked about the cleaning service in the toilets, and 98% of the participants agreed with more regularity of this service in 2020 compared to 92% in 2022. For Q24, the results indicate that in 2020, 98% of people were satisfied with a space ventilated with opened windows and doors where and when possible. Results from 2022 show that 95% of people were satisfied with the measure. Lastly, in Q25, 93% of the respondents agreed with the use of signs and posters to help to reduce the spread of the virus in the first stage, which was very similar to the results from 2022, when 91% of people agreed with the measure. 

### 4.6. Analysis of the Services Dimension

This section shows very interesting results; see Table 7 and Figure 11. Firstly, Q26 expresses the disagreement between people toward changes in the cleaning service. The SD from both stages of the survey are dispersed, so this finding might not be strong enough to make a conclusion but it is relevant considering the very different feelings that people had about the statement. This is also supported by the previous responses from Q7 and Q9 regarding COVID-19 measures that might affect the comfort of respondents. The following questions, Q27 to Q30, seem to show a more positive response in 2020 from the respondents; this is supported by the mode. People seemed to be satisfied and agreed with the hypothetical service conditions exposed, although some of the results have changed in 2022. For instance, results from Q28 have lessened from a mean of 3.35 to a mean of 2.56, and results from Q29 have changed from 4.31 to 3.86. Additionally, Q30 shows a lower mean in 2022 than in 2020 with regard to the level of satisfaction with treatments with minimal contact. Lastly, an important finding has been found from Q31. In this case, people in the first stage presented strong support for the use of technology and smartphones to minimize the interaction when booking a room; respondents strongly agreed with this measure. However, we find a slight decrease in the mean, from 4.38 in 2020 to 3.93 in 2022.

As shown in Figure 12, for Q26, very diverse answers were found. In 2020, 54% of people were not satisfied with the temporary cancellation of the room cleaning service while the guestrooms are occupied, while 23% were neutral about it. However, the number of people not satisfied with this decision increased in 2022 to 68%, and 12% felt neutral about it. Q27 reveals an almost unanimous response with regards to the staff wearing a face mask in 2020; 83% strongly agreed and 16% agreed with this, while in 2022, 53% strongly agreed and 40% agreed. Results from Q28 show that, in 2020, about half of the participants were satisfied or very satisfied with the temporary closing of the restaurants, 25% of them were not sure about it, and 23% disagreed with it. However, in 2022, 56% of people disagreed or strongly disagreed with this decision. Q29 investigates the limit to one guest per booked appointment in the spa. In this hypothesis, 89% of the participants were mostly satisfied with the measure in the first stage but, in the second stage, this was only approved by 76% of respondents. Q30 also shows a decrease from 2020 to 2022 with regard to satisfaction. It was revealed that, in 2020, 77% of people were satisfied or very satisfied; however, in 2022 only 62% responded positively. Finally, Q31 proposes the use of technology for bookings. It can be seen that 89% of the participants from 2020 agreed and strongly agreed with it. A dropping in percentage can be seen in 2022, with 79% of respondents agreeing with the statement.

In summary, based on the 2020 and 2022 surveys, the attributes that received the most positive responses for a sustainable recovery of hotels are, for the aesthetic quality dimension, the style of the ornaments/furniture (Q5, mean 4.30–4.22). For the functionality dimension, the space between the furniture and the use of physical distancing (Q6, mean 4.77–4.31). For the atmosphere dimension, the cleanliness (Q13, mean 4.59–4.61). For the spaciousness dimension, the use of outdoor areas (Q19, mean 4.36–4.32). For the physiological conditions dimension, the air quality (Q24, mean 4.68–4.47). Lastly, for the services dimension, the measures on the staff members (Q27, mean 4.82–4.42).

## 5. Discussion and Sustainable Recovery Attributes

The current study provides a number of interesting findings for practitioners, design experts, and academics of the hospitality and tourism sectors. Table 1 was created in response to the research question RQ1 and, based on the existing literature, in order to present the main servicescape attributes to consider during an airborne health crisis, such as COVID-19. The quantitative analysis presented above shows how the sentiments of people toward servicescapes have changed over the course of the pandemic, with the aim of responding to RQ2. Finally, in order to respond to RQ3, two main groups can be extracted from the results. The first group includes the attributes or improvements that are relevant for people and have remained constant during the phases of the pandemic; these will be called the ‘unaltered attributes’. The second group includes the attributes that have changed from 2020, when the first survey was taken, to 2022, when the second survey was taken and is also relevant; these will be called the ‘altered attributes’. Attributes from both groups must be taken into account and implemented in hotels for a sustainable recovery of the sector.

### 5.1. Unaltered Attributes

The first section, ‘aesthetic quality’, confirmed that customers’ satisfaction increased when the building includes indoor plants. Several recent studies have confirmed the benefits that indoor plants can provide and their contribution to a better indoor environment during COVID-19 [61,62]. Similarly, a minimalist decoration style and clean lines of furniture inspired customers’ positive responses. The perception of the respondents with respect to inhabiting a minimalist hotel is related to the published research of Jiang and Wen (2020), which states that hotel surfaces that receive constant human contact are more likely to be contaminated and become a source of spread of COVID-19 and other infectious diseases. For instance, furniture with clean lines and a minimal design is easier to keep clean. Minimalism has proved to be a sustainable lifestyle that declutters not only spaces but minds. It has also been proved that it alleviates depression and improves wellbeing [63]. People seem to be more comfortable with warning signs in 2022 so this attribute, contrarily to the other COVID-19-related attributes, obtains more positive feedback. Banners and signs about the spread of the virus have proved to be a very efficient method to warn people about it [64]. From the ‘Atmosphere’ section, more than 85% of the participants expressed the importance of natural light and their preference for the use of warm artificial lighting in the room, which is consistent with previous research about the positive effects of natural environments on agitation and stress [55]. The scent of cleanliness was highlighted and, therefore, is essential for questioned people. It has been proven the subconscious positive influence of ambient scents, white bedding, and even the presence of cleaning staff on an individual’s perception of cleanliness during COVID-19 [46]. In regards to the background noise, results confirmed the significance of a quiet and relaxing environment in the hotel. This importance is coherent with several researchers who stated that indoor environmental conditions, such as temperature comfort, lighting, noise or indoor air quality, influence emotional stress and sleeping hours [55]. ‘Spaciousness’ has been a very important factor during COVID-19 and results from this dimension have remained constant from 2020 to 2022. Results clearly show the significance of outdoor space within the hotel. The use of outdoor recreational activity increased by 291% during the lockdown [65] and the value of urban nature during a time of crisis was rediscovered. It has been proved that sometimes, the outdoors helps with mental health and wellbeing during a crisis [39,65]. Participants were also asked about their preferences for having a big guestroom or a big bathroom, and the former was significantly more valuable to them. The importance of indoor air quality has remained constant in both surveys. Airborne viruses and poor indoor air quality are directly related. This is a field that needs to be researched further; so far little research has studied the improvement of air quality within buildings with proper design strategies and the integration of new engineering systems to control it [48]. 

### 5.2. Altered Attributes

The results reveal that most of the participants are aware of the main COVID-19 procedures and measures, although they generally felt more flexible about them in 2022 than in 2020. Physical distancing and the use of masks inside the hotel are very important for the respondents but the mean for both shows that these are not as relevant for them as during the pandemic. Previous research has examined the safety and health measures for COVID-19 in the hospitality industry [66] and found that social distancing in hotels is one of the most effective measures in preventing infections. 

Some of the results from the ‘physiological conditions’ section, provide evidence that most of the respondents agreed with the implementation of all COVID-19 protocols. Although our results clearly show that participants were aware of and understood the restrictions, some of these attributes have been altered from 2020 to 2022. For instance, the use of regular hygiene standards and sanitation facilities is crucial to prevent the spreading of other infectious diseases. People increased their frequency of hand hygiene practices during the pandemic and this should be constant in the future stages as well [67]. 

From the ‘services’ section, a majority of participants strongly agreed with the use of masks by the staff of the hotel. Although this finding has been altered from 2020 to 2022, it is still relevant. This is consistent with published research which demonstrated that most respondents had an acceptable knowledge of the use of face masks and were confident to correctly put it on [68]. The spa services were also examined and respondents supported limited use. This positive attitude toward the use of the spa complements the demonstrated benefits of spas and balneotherapy during COVID-19 [42,47]. The agreement for most of the respondents with the utilization of technology for bookings was revealed. It has been investigated that new uses for technologies, such as the use of live-stream practices, AI, or facial recognition, are utilized on a daily basis to enhance the service quality in the hospitality sector to successfully recover from the virus [45].

In order to facilitate the understanding of the results extracted from the research, a comparison between the means of each survey from the unaltered and altered attributes exposed has been shown in Figure 13.

## 6. Conclusions and Future Research

This paper provides initial insights into the sentiments that people have had toward the hotel servicescapes during and after COVID-19 in a tourism context to support the UNWTO One Planet Vision for a sustainable recovery from the pandemic. Relevant findings can be extracted from this research and the attributes presented should be taken into consideration due to their importance over the years in people’s perspectives. The most relevant finding confirms the importance of greenery; an indoor patio and indoor plants in the hotel are crucial for people’s wellbeing. The second finding suggests that a minimalistic decoration and the incorporation of clean-lined furniture generate a positive response. Practicing minimalism offers significant contributions not only to the body but also to the environment by promoting responsible consumption, a circular economy of products, and a self-sufficient mindset. The third finding confirms the relevance of indoor environmental quality including natural light, warm artificial lighting, a relaxing/quiet background, controlled air quality, and the scent of cleanliness proves to be beneficial. The fourth finding suggests that the feeling of indoor and outdoor spaciousness have a very positive impact on people. Lastly, participants seem to feel more flexible in the second stage of the post-pandemic era regarding COVID-19 measures. These results were expected, considering that a high number of the population are now vaccinated and governments have eased the restrictions. However, people emphasized the importance of physical distancing, facial masks, limited use of the spa, and online bookings in the hotel to cope with the virus, even in 2022. It was noticeable that the majority of the respondents have a clear knowledge of the recommended procedures and feel cooperative; however, results also suggest that respondents were slightly reluctant when these actions affect their comfort. Additionally, technological innovation can play a key role to reduce customers’ perceived health risks. From a broader perspective, the physiological conditions and atmosphere dimensions have the highest mean value, and positive responses from the participants about the servicescapes of hotels during and following the pandemic. This is an important finding and shows the dimensions that positively impact a sustainable recovery of the sector in the post-pandemic era.

The limitations of our survey are common to most surveys involving personal responses. For instance, the validity of the Likert scale attitude measurement can be compromised due to social desirability; this means that the individuals involved in the survey may lie to present themselves in a positive light. In this case, the questionnaire asked relevant questions with regard to COVID-19 measures, and although most of the respondents agreed with these restrictions, there is a chance that they might not be truthful with their statements and follow instead what is socially accepted.

The magnitude of the COVID-19 crisis represents a challenge for all researchers of different disciplines. The pandemic has paralyzed the course of hospitality toward zero neutrality, but adequate management is crucial for a sustainable recovery and to meet future sustainable plans. Understanding what needs to be improved, what are people’s needs, and what are the right tools to use are priorities from now to 2050. Theoretical implications have been extracted from this research, and the main aspects that should be deeply analyzed and incorporated in hotels have been presented to cast some light on the impact that COVID-19 has had upon people’s wellbeing in a hotel. Additionally, practical implications can be drawn for hospitality managers and stakeholders of hotels to understand their customer’s needs and feelings. It is time for both the hospitality industry and scholars to work together and bring new models, approaches, and ideas that will help to overcome the devastating effects of the pandemic. 

Future research will be required to provide further data for a sustainable recovery of the sector in the post-pandemic era. For instance, the spaciousness and atmosphere dimensions have not rejected the null hypothesis presented as their *p*-value, which is higher than the significance considered; as a consequence, it has not been possible to prove a significant effect on these dimensions. Although some of their attributes have shown a positive response, other attributes have remained unaltered and do not demonstrate a significant difference between 2020 and 2022. Further research about these dimensions might be needed to reinforce the unaltered nature of these attributes.

## Figures and Tables

**Figure 1 ijerph-20-01100-f001:**
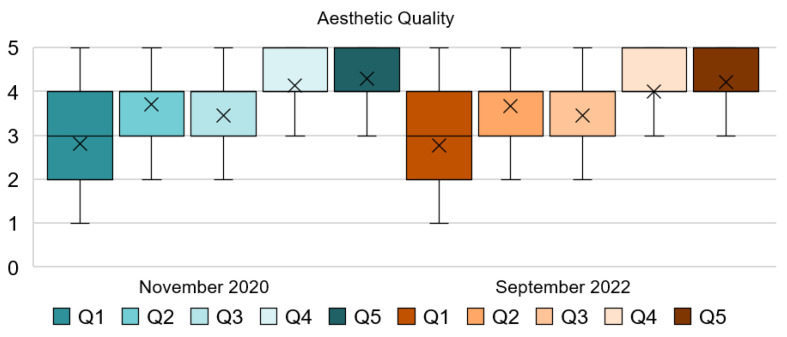
Interval data graph of the Aesthetic Quality dimension.

**Figure 2 ijerph-20-01100-f002:**
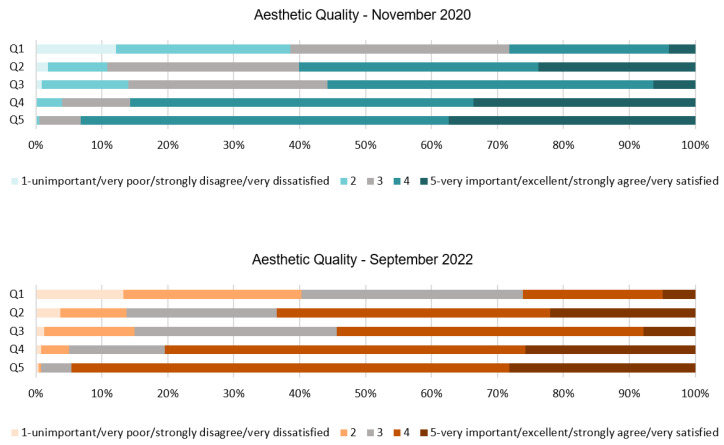
Aesthetic Quality dimension chart.

**Figure 3 ijerph-20-01100-f003:**
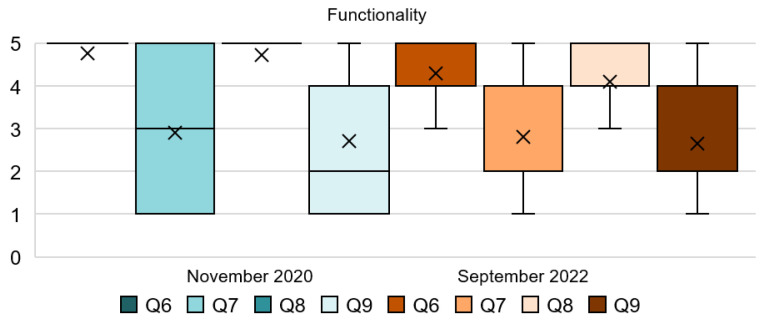
Interval data graph of the Functionality dimension.

**Figure 4 ijerph-20-01100-f004:**
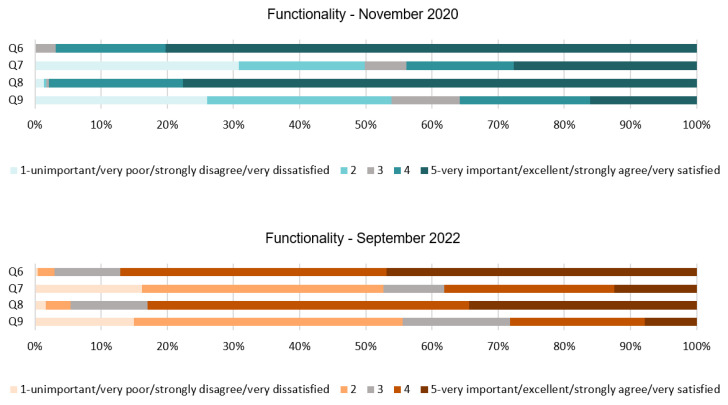
Functionality dimension chart.

**Figure 5 ijerph-20-01100-f005:**
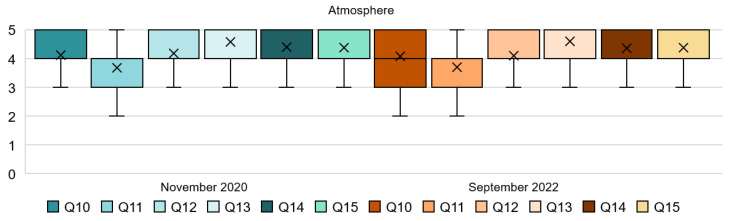
Interval data graph of the Atmosphere dimension.

**Figure 6 ijerph-20-01100-f006:**
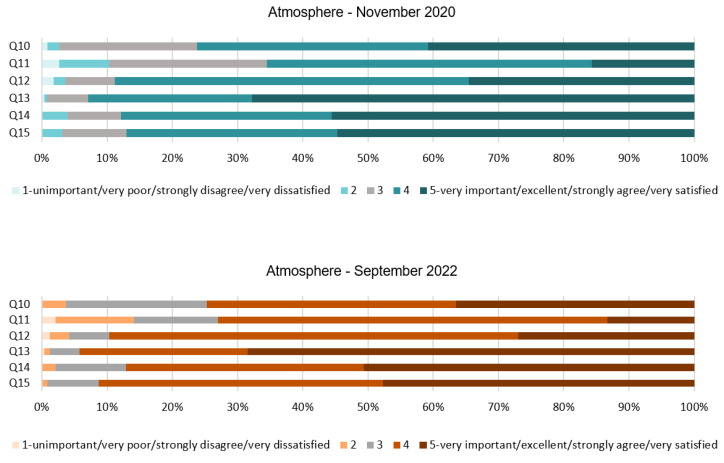
Atmosphere dimension chart.

**Figure 7 ijerph-20-01100-f007:**
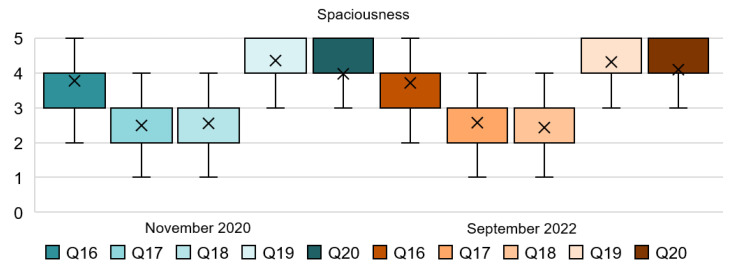
Interval data graph of the Spaciousness dimension.

**Figure 8 ijerph-20-01100-f008:**
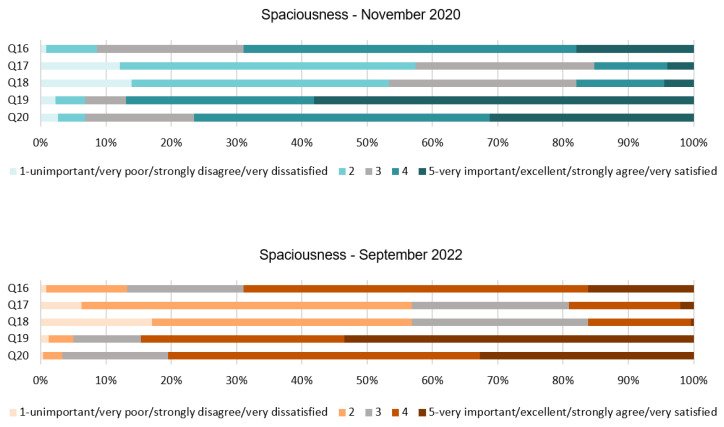
Spaciousness dimension chart.

**Figure 9 ijerph-20-01100-f009:**
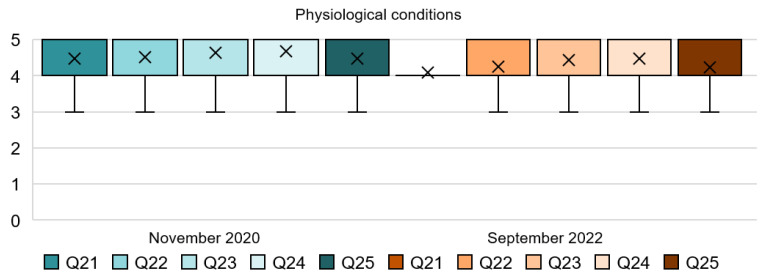
Interval data graph of the Physiological Conditions dimension.

**Figure 10 ijerph-20-01100-f010:**
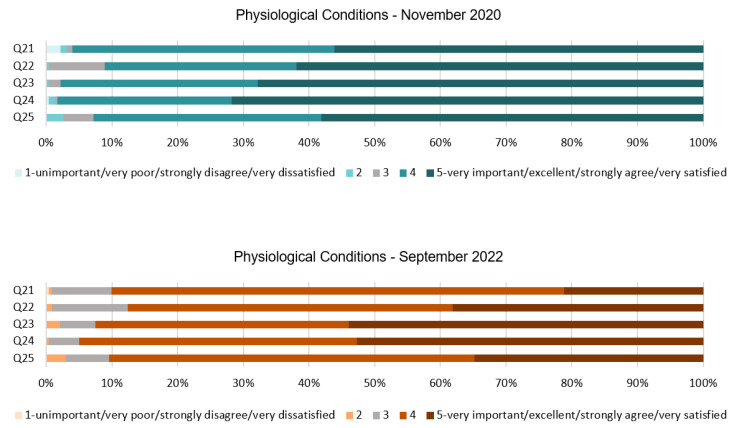
Physiological Conditions dimension chart.

**Figure 11 ijerph-20-01100-f011:**
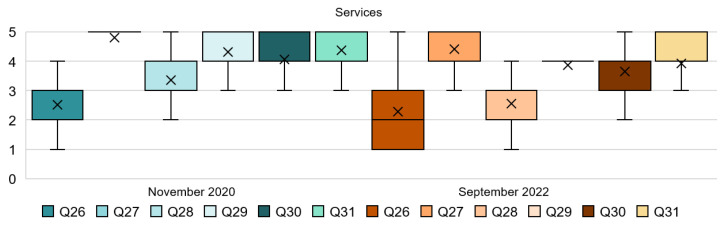
Interval data graph of the Services dimension.

**Figure 12 ijerph-20-01100-f012:**
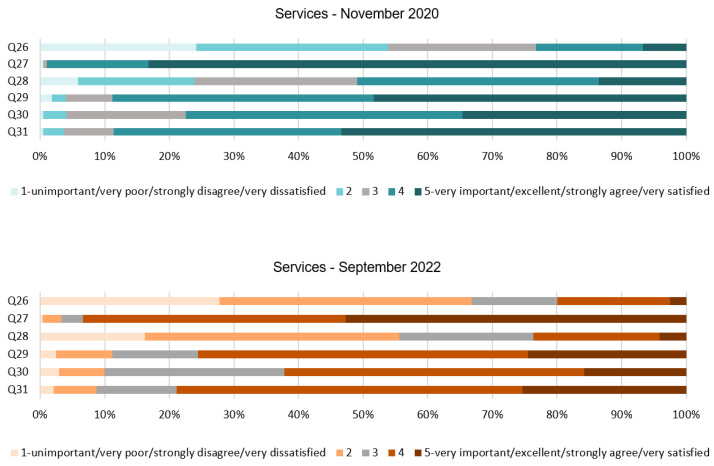
Services dimension chart.

**Figure 13 ijerph-20-01100-f013:**
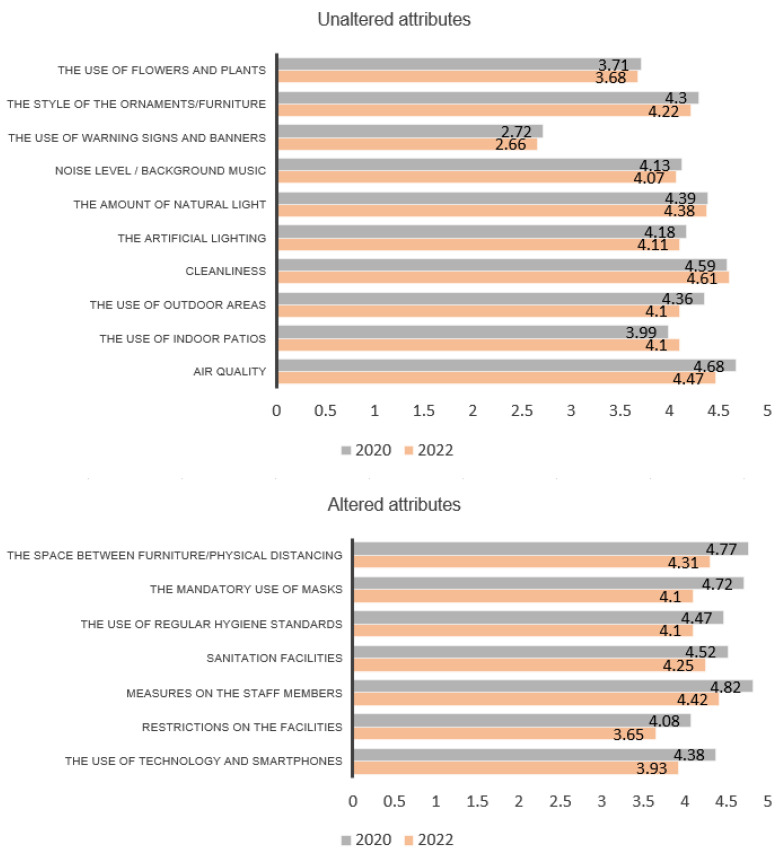
Unaltered and altered attributes for a sustainable recovery.

**Table 1 ijerph-20-01100-t001:** The servicescape scale during an airborne health crisis developed by the authors.

Dimensions	Attributes	Servicescape References	COVID-19 References
Aesthetic Quality	The pictures and photos on display	Bitner (1992); Lockwood and Pyun (2018)	Keenan (2020)
	The style of the ornaments	Bitner (1992); Wakefield and Blodgett (1994); Lockwood and Pyun (2018); Baker (1986)	Keenan (2020); Jiang and Wen (2020); Shamaileh (2021)
	The use of flowers and plants	Lockwood and Pyun (2018)	Jackson et al. (2021); Jeon and Yang (2021)
	The style of the furniture used	Bitner (1992); Wakefield and Blodgett (1994); Lockwood and Pyun (2018); Baker (1986)	Keenan (2020); Jiang and Wen (2020); Shamaileh (2021)
	The flooring design	Wakefield and Blodgett (1994); Lockwood and Pyun (2018); Baker (1986); Lugosi et al. (2022)	Keenan (2020); Ekwall et al. (2021)
Functionality	The space between furniture	Bitner (1992); Wakefield and Blodgett (1994); Lockwood and Pyun (2018); Baker (1986); Lugosi et al. (2022)	Shin and Kang, 2020; Zhao et al., 2020
	The practicality of the flooring	Lockwood and Pyun (2018); Lugosi et al. (2022)	Keenan (2020); Ekwall et al. (2021)
	The use of physical distancing *	Proposed by the authors	Shin and Kang (2020); Zhao et al. (2020)
	The use of warning signs and banners	Bitner (1992); Baker (1986)	Ekwall et al. (2021)
	The mandatory use of masks *	Proposed by the authors	Zhai (2020); Blocken et al. (2021)
Atmosphere	The artificial lighting	Lockwood and Pyun (2018); Ozkul et al. (2019); Lugosi et al. (2022)	Miao and Ding (2020)
	Background music	Bitner (1992); Lockwood and Pyun (2018)	Hennessy et al. (2021); Ziv and Hollander-Shabtai (2021); Zarrabi et al. (2021)
	The amount of natural light	Lockwood and Pyun (2018); Lugosi et al. (2022)	Miao and Ding (2020); Zarrabi et al. (2021)
	Noise level	Bitner (1992); Lockwood and Pyun (2018); Baker (1986)	Miao and Ding (2020); Zarrabi et al. (2021)
	Cleanliness	Bitner (1992); Wakefield and Blodgett (1994); Lockwood and Pyun (2018);	Vilnai-Yavetz and Gilboa (2010); Shin and Kang (2020); Magnini and Zehrer (2021)
Spaciousness	The amount of free space and the feeling of spaciousness	Lockwood and Pyun (2018); Lugosi et al. (2022)	Keenan (2020); Jeon and Yang (2021); Jackson et al. (2021); Zarrabi et al. (2021)
	The use of outdoor areas *	Proposed by the authors	Jeon and Yang (2021); Jackson et al. (2021)
	The use of indoor patios *	Proposed by the authors	Jackson et al. (2021)
Physiological Conditions	Sanitation facilities *	Proposed by the authors	Vilnai-Yavetz and Gilboa (2010); Shin and Kang (2020); Girard et al. (2019)
	The use of regular hygiene standards *	Proposed by the authors	Vilnai-Yavetz and Gilboa (2010); Shin and Kang (2020); Girard et al. (2019)
	Air quality	Bitner (1992); Baker (1986)	Blocken et al. (2021); Megahed and Ghoneim (2021)
Services	Cleaning service	Wakefield and Blodgett (1994); Baker (1986)	Vilnai-Yavetz and Gilboa (2010)
	Measures on the staff members *	Proposed by the authors	Zhai (2020); Savavibool (2016)
	Restrictions on the facilities *	Proposed by the authors	Kardeş (2021); Martínez-Moure and Saz-Peiró (2021)
	The use of technology and smartphones	Wakefield and Blodgett (1994); Willems et al. (2021)	García et al. (2021); Sztorc (2022); Xiang et al. (2022); Lau (2020)

* These attributes have been proposed by the authors and are based on the existing research available.

**Table 2 ijerph-20-01100-t002:** Interval data analysis of the Aesthetic Quality dimension.

Aesthetic Quality	t: 2.50	*p*-Value: 0.03					
Item	Attributes Related to the Question	November 2020			September 2022		
		Mode	Mean	SD	Mode	Mean	SD
Q1. How important is it for your comfort that there is the use of artwork in the communal areas of the hotel?	The pictures and photos on display	3	2.82	1.06	3	2.78	1.08
Q2. Is it important for you to find indoor plants in the building?	The use of flowers and plants	4	3.71	0.99	4	3.68	1.04
Q3. How do you feel in the room if there are no indoor plants?	The use of flowers and plants	4	3.47	0.83	4	3.46	0.87
Q4. A minimal space is easier to keep clean and disinfected. Do you feel comfortable with a minimalist decoration style?	The style of the ornaments The style of the furniture used The flooring design	4	4.15	0.76	4	4	0.80
Q5. How do you feel about a room with clean lines of furniture?	The style of the ornaments The style of the furniture used	4	4.30	0.61	4	4.22	0.58

**Table 3 ijerph-20-01100-t003:** Interval data analysis of the Functionality dimension.

Functionality	t: 2.22	*p*-Value: 0.05					
Item	Attributes Related to the Question	November 2020			September 2022		
		Mode	Mean	SD	Mode	Mean	SD
Q6. How important is it for the implementation of physical distancing in the hotel during the pandemic?	The space between furniture The use of physical distancing	5	4.77	0.49	5	4.31	0.78
Q7. A lobby with furniture keeping the recommended distance to prevent the spread of the virus makes you feel uncomfortable.	The space between furniture The use of physical distancing	1	2.91	1.64	2	2.82	1.32
Q8. Would you agree with the use of a mask for the hotel staff?	The mandatory use of masks	5	4.72	0.63	4	4.10	0.87
Q9. The use of floor stickers, signs, and banners supports a safe environment in the hotel. Do you think these measures affect your comfort?	The use of warning signs and banners	2	2.72	1.45	2	2.66	1.19

**Table 4 ijerph-20-01100-t004:** Interval data analysis of the Atmosphere dimension.

Atmosphere	t: 1.37	*p*-Value: 0.11					
Item	Attributes Related to the Question	November 2020			September 2022		
		Mode	Mean	SD	Mode	Mean	SD
Q10. How important is for a quiet environment?	Noise level Background music	5	4.13	0.87	4	4.07	0.85
Q11. Chill-out music would make you feel more comfortable.	Background music	4	3.68	0.92	4	3.70	0.92
Q12. Does the use of warm lighting in your room make you feel calm?	The artificial lighting	4	4.18	0.79	4	4.11	0.74
Q13. Cleanliness is key for reducing the risk of spreading the virus, but how important is it for you a to have a clean scent in the hotel?	Cleanliness	5	4.59	0.67	5	4.61	0.66
Q14. Due to the amount of time that you might spend in the hotel, how important is it for you to have natural light in the lobby?	The amount of natural light	5	4.39	0.80	5	4.36	0.76
Q15. How important is it for you to have natural light in the restaurant?	The amount of natural light	5	4.39	0.79	5	4.38	0.67

**Table 5 ijerph-20-01100-t005:** Interval data analysis of the Spaciousness dimension.

Spaciousness	t: 0.90	*p*-Value: 0.20					
Item	Attributes Related to the Question	November 2020			September 2022		
		Mode	Mean	SD	Mode	Mean	SD
Q16. A big hotel guestroom is more important for you than a big bathroom.	The amount of free space and the feeling of spaciousness	4	3.78	0.87	4	3.71	0.91
Q17. A big bathroom is more important for you than a big hotel guestroom.	The amount of free space and the feeling of spaciousness	2	2.50	0.98	2	2.58	0.91
Q18. Due to the amount of time that you might spend in the hotel, how important is it for you to have a double-height space in the lobby?	The amount of free space and the feeling of spaciousness	2	2.55	1.03	2	2.43	0.96
Q19. How important is it for you to find an outdoor common area in the hotel?	The use of outdoor areas	5	4.36	0.95	5	4.32	0.90
Q20. Considering the amount of time you might spend in the hotel, an indoor patio would make you feel more relaxed and less overwhelmed.	The use of indoor patios	4	3.99	0.94	4	4.10	0.80

**Table 6 ijerph-20-01100-t006:** Interval data analysis of the Physiological Conditions dimension.

Physiological Conditions	t: 2.16	*p*-Value: 0.04					
Item	Attributes Related to the Question	November 2020			September 2022		
		Mode	Mean	SD	Mode	Mean	SD
Q21. Hotels might need to provide regular reminders and signage to maintain hygiene standards. Would you be satisfied with this measure?	The use of regular hygiene standards	5	4.47	0.77	4	4.10	0.59
Q22. How important is it for you to stay in a hotel with enough sanitation facilities?	The use of regular hygiene standards Sanitation facilities	5	4.52	0.67	4	4.25	0.69
Q23. Keeping the toilets clean after your use is also important to reduce the spread of the virus. Would you agree with more regularity of the cleaning service during your stay in the hotel?	The use of regular hygiene standards	5	4.65	0.57	5	4.44	0.69
Q24. Keeping a space ventilated by opening windows and doors where and when possible reduces the risk of spreading the virus. Would you be satisfied with this measure?	Air quality	5	4.68	0.58	5	4.47	0.61
Q25. Using signs and posters to build awareness of good handwashing techniques might be helpful to reduce the spread of the virus. Would you agree with this measure?	The use of regular hygiene standards	5	4.48	0.71	4	4.22	0.69

**Table 7 ijerph-20-01100-t007:** Interval data analysis of the Services dimension.

Services	t: 6.25	*p*-Value: 0.0007					
Item	Attributes Related to the Question	November 2020			September 2022		
		Mode	Mean	SD	Mode	Mean	SD
Q26. Some hotels have temporarily canceled the room cleaning service while guestrooms are occupied. Would you be satisfied with this decision?	Cleaning service	2	2.52	1.21	2	2.28	1.12
Q27. Staff might be wearing a face mask to reduce the risk of spreading the virus. Would you approve of this measure?	Measures on the staff members	5	4.82	0.46	5	4.42	0.73
Q28. Some hotels have temporarily closed all the restaurant areas and just offered room service. Would you be satisfied with this decision?	Restrictions on the facilities	4	3.35	1.10	2	2.56	1.10
Q29. The spa management might consider limiting its use to one guest per booked appointment with a break between sessions for cleaning. Would you be satisfied with this decision?	Restrictions on the facilities	5	4.31	0.84	4	3.86	0.97
Q30. Treatments with minimal contact are recommended for guests that may feel nervous about visiting the spa. Would you be satisfied with this choice?	Measures on the staff members Restrictions on the facilities	4	4.08	0.84	4	3.65	0.93
Q31. The use of technology such as smartphones, tablets, or laptops for bookings minimizes the interaction between staff and guests and helps to reduce the risk of spreading the virus. Do you agree with this alternative?	The use of technology and smartphones	5	4.38	0.80	4	3.93	0.91

## Data Availability

The data presented in this study are available on request from the corresponding author. The data are not publicly available to keep the privacy of all participants.

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
