# Peer review of "Promoting the Sustainable Recovery of Hospitality in the Post-Pandemic Era: A Comparative Study to Optimize the Servicescapes"

_ijerph, 2023, doi:10.3390/ijerph20021100_

Round 1

Reviewer 1 Report

1. Did the author conduct a hypothesis test which shows that there is a significant effect of each dimension on customer satisfaction? several descriptions of the discussion sentences indicate a significant statement.

2. For  RQ3. What are the attributes that must be taken into account for a sustainable recovery  of hotels? authors can rank on each dimension such as  Aesthetic Quality : The style of the ornaments The style of the furniture used  (Q5), Functionality dimension : The space between furniture  & The use of physical distancing. (Q6), Atmosphere dimension : Cleanliness  (Q13)  , Spaciousness dimension (Q19) The use of outdoor areas ,  Physiological Conditions dimension  ; Air Quality (Q24) , Services dimension : Measures on the staff members (q27) , the most important Attributes in servicescape. 

as well as comparing the mean values of unaltered and altered attributes  , so that it is easier for readers to read the results of their research. although it has been well explained in the Discussion 

  Nov-20 Sep-22

Dimension Mean Mean

Aesthetic Quality  3.69 3.63

Functionality 3.78  3.57

Atmosphere 4.23         4.21

Spaciousness 3.44        3.43

Physiological Conditions  4.64 4.29

Services          3.91 3.45

 based on the mean value, Physiological Conditions and Atmosphere is the highest dimension considered by respondents on the characteristics of the hotels. maybe this could be an important finding in this paper

3. The author might be able to add a future research direction at the end of the conclusion

Author Response

The authors appreciate the comments of the reviewer that have helped to improve the manuscript. A new version of the manuscript is available, and some answers to the key comments can be found below.

Point 1: Did the author conduct a hypothesis test which shows that there is a significant effect of each dimension on customer satisfaction? several descriptions of the discussion sentences indicate a significant statement.

Response 1: Thank you for your kind words in favour of our research. All the comments and recommendations provided have been extremely useful for this paper. A t-test was conducted to provide the data presented in the document, however the t and p values were not included. A further explanation about the statistical test conducted in this research to study the significance of each dimension have been included and can be found from lines 235-240. The significance level for the research is shown in line 231. In addition, the t-value and p-value have been added for each dimension at the top of tables 2-7. With these values, the null hypothesis remains or is rejected. We hope that with this information provided, the study is clearly presented and statistically analysed.

Point 2: For RQ3. What are the attributes that must be taken into account for a sustainable recovery of hotels? authors can rank on each dimension such as Aesthetic Quality: The style of the ornaments The style of the furniture used (Q5), Functionality dimension: The space between furniture & The use of physical distancing. (Q6), Atmosphere dimension: Cleanliness (Q13), Spaciousness dimension (Q19) The use of outdoor areas, Physiological Conditions dimension; Air Quality (Q24), Services dimension: Measures on the staff members (Q27), the most important Attributes in servicescape. 

Response 2: Thank you for this recommendations, it truly summarises the results from the research very easily for readers. These, among the mean values, have been included at the end of the results section from lines 471-478.

Point 3: As well as comparing the mean values of unaltered and altered attributes so that it is easier for readers to read the results of their research, although it has been well explained in the Discussion.

Response 3: Thank you for this useful suggestion, this will indeed help to understand how the article is structured. In order to achieve this, we have included Figure 13 together with a brief explanation from lines 570-572. Figure 13 shows a comparison between the highest means of surveys 2020 and 2022 for both categories of attributes, unaltered and altered. Hopefully this new figure will help the readers to comprehends the most relevant attributes for a sustainable recovery and how these attributes have changed from the beginning of the pandemic to the current Post-Pandemic stage.

Point 4: Based on the mean value, Physiological Conditions and Atmosphere is the highest dimension considered by respondents on the characteristics of the hotels. Maybe this could be an important finding in this paper

Response 4: We really appreciate this suggestion. Although the attributes have been exposed individually, a broader perspective about the dimensions will highlight the most relevant dimensions to consider. Hence, this important finding has been included from lines 598-602.

Point 5: The author might be able to add a future research direction at the end of the conclusion.

Response 5: Thank you for this suggestion in order to enhance the final part of the document. Indeed, future research will complete the conclusions section and provide a new path for future researchers to follow in this topic. As a consequence, a considering that two of the dimensions have not been altered from 2020 to 2022, further research about this aspect might be needed to confirm this or reject otherwise these hypotheses. This can be found from lines 622-629.

We would like to thank the reviewers for their assistance, suggestions and ideas. Hopefully, the changes that have been exposed above will help to enhance the manuscript to make it more clear for the future readers and researchers of the sustainable recovery of buildings in the Post-Pandemic Era.

Reviewer 2 Report

Dear respected authors, 

Regarding Sustainable recovery, the topic must be explained more by citing relevant references.

About Service space effects, referred references are conducted in 92,94... is there any new ones? Please support the idea with the latest publications.

In Table 1, it is not clear how you reached the categories at all!!

In conclusion from the beginning, you mentioned some items, first, second and third... while none of them is related to COVID-19 or the post-pandemic era.  Indeed, it is the main weakness of your research. The obtained results are valid in the normal situation, and not related to the pandemic and post-pandemic era. 

Author Response

The authors appreciate the comments of the reviewer that have helped to improve the manuscript. A new version of the manuscript is available, and some answers to the key comments can be found below.

Point 1: Regarding Sustainable recovery, the topic must be explained more by citing relevant references.

Response 1: Thank you for your kind review and assistance in favour of our research. Section 2.1. of the manuscript, exposes a literature review about the “Implications of COVID-19 in the sustainable development of hospitality”. This section analyses the current status of the sector in the Post-Pandemic Era, however, we agree with the reviewer that further explanation about the sustainable recovery of the sector in the Post-Pandemic Era will help to enhance the manuscript. For instance, the relation between tourism and the 2030 Agenda, together with the sustainable path towards 2050 neutrality have been explained further from lines 80-86. Additionally, relevant recent references about the effects of the pandemic towards sustainable development have been included from lines 104-115. We hope that these improvements will help readers and researchers to better understand the current context of the research.

Point 2: About Service space effects, referred references are conducted in 92,94... is there any new ones? Please support the idea with the latest publications.

Response 2: Thank you for this useful suggestion, the addition of latest publications will indeed help to support the topic of the research. The references from 1992 and 1994 correspond to the creators of the concept of “servicescape”. These references have helped to shape the current research and have been used as a based to upgrade the proposed scale. However, we strongly agree with the reviewer that the addition of latest publications in regards of servicescape, will help to support the main idea of the research. In connection with this, recent and relevant references have been included in lines 135-137 and lines 146-160. Additionally, these references have helped to complete Table 1 since the column of “servicescape” was also missing some important latest publications. We hope you find this improvement useful.

Point 3: In Table 1, it is not clear how you reached the categories at all.

Response 3: Thank you for this useful suggestion, indeed further explanation about the design of the proposed servicescape scale will help others to understand its relevance and structure. Consequently, the paragraph shown in lines 192-199 has been added. It explains the meaning of each column and how the proposed scale has been designed. Existing scales were too generic to be used during or after the pandemic and did not focus on any specific context. The scale presented in this research provides an upgraded list of attributes and dimensions of the servicescape to be used in a pandemic situation. The scale is supported by existing literature and recent publications and it has been created as a tool for future researchers of the field. Hopefully with the further explanation included in the manuscript, your suggestion has been fulfilled.

Point 4: In conclusion from the beginning, you mentioned some items, first, second and third... while none of them is related to COVID-19 or the post-pandemic era.  Indeed, it is the main weakness of your research. The obtained results are valid in the normal situation, and not related to the pandemic and post-pandemic era. 

Response 4: We really appreciate this suggestion and we would like to explain further how the research questions have been designed and their main purpose. Although the entire research has been designed to promote a sustainable recovery of the sector in the Post-Pandemic Era, we agree that the research questions did not clearly stated the nature of the research. As a consequence, these have been rewritten to clearly show the main purpose of the methodology.

On a side note, we would like to highlight that the entire methodology is focussed on the Post-Pandemic context, including the methodology and results. For instance, the proposed servicescape scale has been designed from the beginning to show the attributes that can be applied in a pandemic environment. A sixth dimension named “Services” has been also included, and it is purely focussed on a pandemic scenario. The attributes that have been proposed on previous scales by other authors, but are not supported by existing research and latest publications in a pandemic context, have been omitted from our proposed scale. This has been explained in lines 199-203. Moreover, the questionnaire is based on the proposed scale and it has been entirely designed from a pandemic point of view. Participants were asked to respond the questions/statements considering the time they were in at the time, this is November 2020 and September 2022. This last information about the participants was not presented in the first version of the paper and it has been added in lines 171-172 in the current version for a better understanding of how the questionnaire was answered. The questionnaire was administered during the pandemic and after it, so its context clearly shows the significance of the paper and its relation with the Post-Pandemic Era.

Finally, results extracted from the methodology and the discussion sections present the most relevant attributes to be considered at the current Post-Pandemic Era for a sustainable recovery and to achieve the 2050 neutrality. Figure 13 has been included to highlight the most relevant attributes that must be considered for a sustainable recovery in the hospitality sector at the Post-Pandemic Era. Although some of the attributes might be also applied in a non-pandemic context, both unaltered and altered attributes have acquired more relevance for people due to the pandemic, as it can be seen in the mean value and, as a consequence, they must be prioritised to promote a healthy recovery. Other attributes, such as the mandatory use of masks, the use of regular hygiene standards and sanitation facilities among others, have been particularly included for a pandemic context so their relationship with the Post-Pandemic Era is inevitable.

At the moment, there is very limited research about the sustainable recovery of the sector from a servicescape perspective in the Post-Pandemic Era. The One Planet vision has been designed by the UNWTO as a response for a sustainable recovery of the tourism sector but there are none publications that support this sustainable plan. We hope that our paper provides some important insights and data that support the One Planet vision and that is the main goal of this research.

We would like to thank the reviewers for their assistance, suggestions and ideas. Hopefully, the changes that have been exposed above will help to enhance the manuscript to make it more clear for the future readers and researchers of the sustainable recovery of buildings in the Post-Pandemic Era.

Round 2

Reviewer 2 Report

Thanks for considering the comments.